# China in Africa: An Examination of the Impact of China's Loans on Growth in Selected African States

Courage Mlambo 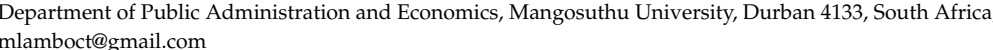

Department of Public Administration and Economics, Mangosuthu University, Durban 4133, South Africa;
mlamboct@gmail.com

**Abstract:** This study sought to test the impact of China's infrastructure investment on economic growth in selected African states. Many comparative studies have shown the positive role that infrastructural loans plays in supporting economic growth. However, for Africa, the role of China's infrastructure projects has mixed views with regards to its contribution to growth and development. A survey of the literature showed that the central question about Chinese infrastructural loans is whether the infrastructural projects are beneficial or detrimental to Africa. Currently, there is no settled opinion as to whether (or not) Africa is benefiting from the Chinese economic relations. This study was quantitative, and we used panel data to achieve our objectives. The study employed annual panel data for 15 African countries covering the period of 2000–2017. The Pooled Mean Group, Mean Group, Fully Modified Ordinary Least Squares, and Dynamic Ordinary Least Squares panel techniques were used for estimation purposes. The main conclusion from the quantitative analysis of China's infrastructural loans in Africa is that China's efforts in developing infrastructure are translating into economic growth. This study provides evidence that China's engagement in Africa could be beneficial, given the positive relationship between loans and economic growth.

**Keywords:** China-Africa relations; infrastructure development; loans; debt; debt overhang; debt trap diplomacy





## 1. Introduction and Background of Study

This study seeks to examine the impact of China's infrastructure investments on economic growth in selected African states. Chinese development finance has significantly risen in recent years, and is now one of the most essential sources of development finance in Africa. Through its Belt and Road strategy, arrangements with the African Union, and at the forum on China–Africa cooperation, the Chinese extended their "win–win" economic policy by investing in socio-development projects (Jayaram et al. 2017; Dahir 2019).

Between 2000 and 2018, China provided $148 billion of loans to African countries (Olander 2020; Sun 2020; Wheatley et al. 2020), and this has made China Africa's biggest bilateral lender (Preuss 2021; Subacchi 2021). African countries faced with a desperate need to boast their infrastructure, development, and economic growth have increasingly turned to China for development finance. China's loans to African countries have the capability to promote economic growth and development.

Nonetheless, China's development finance activities to Africa have come under scrutiny in recent years (Humphrey and Michaelowa 2019; Procopio 2020; Stones and Chazan 2020), with critics noting that China's loans could encourage enslavement, entrap African countries in debt, and drive debts to unsustainable levels (Moore 2018; Were 2018; Stones and Chazan 2020). Grant Harris, the top Africa diplomat in former US president Barack Obama's administration, described Chinese loans as "the methamphetamines of infrastructure finance: highly addictive, readily available, and with long-term negative effects that far outweigh any temporary high" (Sautman 2019).

The magnitude of China's lending to African countries as well as China's motives has been labelled as "debt trap diplomacy". China's debt trap diplomacy is often labelled as a long-term aim to trap African countries in debt commitments they cannot fulfil and eventually leverage that pretext to obtain African resources or important assets (Kazeem 2020). China's activities in Africa have been questioned by, mainly, Western nations who believe that China impedes Africa's growth (Sautman 2019; Feng and Pilling 2019). China's financing is painted as predatory (Gopaldas 2018), and China is also blamed of offering unsustainable loans for inefficient infrastructure schemes in Africa to ensnare them in debt.

It is also believed that the implications of China's lending actions increase the credit risks facing African countries (Rogovic and Robinson 2018). Mihalyi cited in Zimbabwe Coalition of Debt and Development (2021) argued that, even though these loans have often provided much-needed infrastructure, such as roads and hydro-dams, in several cases, they have led to increasing and unsustainable levels of debt and the danger of losing collateral is worth more than the value of the loan. China is also blamed for providing loans to projects with a weak linkage to growth.

For example, China–Africa infrastructure development partnership are sometimes seen to be constructing vanity projects. China is blamed for offering loans that fund the construction of government facilities, cultural centres, and sports facilities that offer limited or temporary economic benefits (Will 2012; Strange et al. 2013; Rosen 2018). It can thus be said that China's development finance may not contribute to economic growth if it is funding vanity projects. Except the loans provided by China create considerable economic benefits improving debt servicing capability, the loans will have substantial negative consequences for development and debt sustainability.

While critics are concerned about China's interest in Africa, some African leaders are embracing and praising the loans from China. Eleanor (2017) concurred and stated that many African leaders have praised the benefits of Chinese loans to promote growth in their countries. Some African leaders believe that China's infrastructure model does not involve many hurdles as China does not place any strings to its projects (Eleanor 2017). Furthermore, although there have been concerns about China's involvement in Africa, neither the U.S. nor its richest allies have proposed anything of substance to compete with China over the basic infrastructure so vitally needed by African countries (French 2021).

This makes China's model attractive to Africa. Part of the desirability of Chinese development finance is that the loans are provided as subsidised and have long repayment periods and relatively low interest rates (Bräutigam 2018; Edinger and Labuschagne 2019). Furthermore, African countries have turned to China because of its "no conditionality policy" (Kwasi 2019). China's quick assistance is attractive because traditional funders, such as Western countries and institutions, scaled down their supply of development finance to Africa (Baker McKenzie 2018; Kwasi 2019).

When the above information is taken into consideration, it can be realized that there is no settled opinion with regards to the impact of China's infrastructure loans on growth in Africa. In order to dismiss or confirm these accounts, there is a need to take a critical evaluation, review, and investigation of Africa's relationship with China. This study sought to answer the following question: do China's loans have an impact on growth in Africa?

To the best of this study's knowledge, there was no prior study conducted to econometrically examine the role of infrastructural loans on the economic growth of African countries. Unfortunately, the area is not very well-researched and has been relegated to the realms of newspaper articles. Stein and Uddhammar (2021) concurred and stated that the China–Africa engagement debate rests upon anecdotal accounts in media reports, cherry-picked cases, and isolated excerpts. This study, therefore, makes an original contribution towards the scope of the China–Africa relations. The paper empirically tests the impact of China's infrastructure investment on economic growth in Africa.

The paper is divided into six sections, following the introduction in the Section 1, Section 2 presents a review of literature. Section 3 provides a review of the dynamics of the

China–Africa loans. Section 4 presents the methodology that was utilised to conduct the study, whist Sections 5 and 6 discuss the findings and conclusion of the study, respectively.

## 2. Literature Review

### 2.1. Theoretical Literature

In order to expedite economic growth, African nations frequently rely on foreign sources to fill local investable capital limitations caused by low domestic savings. There are several theories that explain the advantages and disadvantages of relying and excessively depending on foreign loans. The theories are discussed below.

### 2.1.1. Financing Gap Theory (Two Gap Theory)

Chenery and Strout (1966) highlighted that foreign aid could be used to fill either a savings gap or a foreign change gap needed for less developed countries' economic development. This model is called the "Two-gap model" and has been discussed extensively in the context of foreign aid effectiveness. Chenery and Strout (1966) argued that foreign loans can be used to replenish national savings in the case of a temporary mismatch between investment and saving capabilities. A country can only self-refinancing when its marginal saving rate is high enough.

When the savings levels are low, a country will have to source from foreign sources. According to some experts, the model has proven to be one of the most widely utilized theories in economics (Efendi 2001) and is equally used by International Finance Institutions (IFIs) in attaining all funding required decisions (Easterly 1999). This theory is applicable to this study because African countries do not have funds to finance infrastructure projects. Throughout the years, they have been relying on foreign assistance to boast their economies. It can thus be said that China's willingness to give loans to African countries might be considered a significant windfall to the continent. China's infrastructural funds could finance the infrastructure gap that Africa is currently facing.

### 2.1.2. False Paradigm Theory

This is premised on the belief that underdeveloped nations (particularly African countries) have failed to progress because their development methods, which are typically provided by foreign ethnocentric economists from wealthy countries, are based on a flawed development model. While many multilateral organizations (World Bank and IMF) from wealthy nations may give well-intentioned and beneficial advice and finance, the False Paradigm contends that they may not have the correct viewpoint when dealing with difficulties of economic growth in developing countries (Jouanjean and Velde 2013).

Western countries and institutions, such as the World Bank and IMF have often been blamed for attaching political conditions whenever they provide African countries with loans. Western countries and institutions require the recipient to undergo changes in governance structure and adhere to international best practices (Li 2018). This has been blamed for causing underdevelopment in African countries. On the other hand, China does not attach strings or conditions, and this has led to African countries embracing China's engagement (Li 2018; Wale 2018).

It can thus be said that China's infrastructural loans could promote growth and development because they do not have political conditions. However, Africa's traditional partners, particularly the United States and Europe, have questioned the reasons and motive of Chinese loans in Africa on several occasions (Moore 2021). They have argued that China is saddling Africa with debt. Some analysts, however, claim that Western countries have exaggerated the China–Africa relationship (Kazeem 2020 Chelwa cited in Acker 2021).

### 2.1.3. Debt Overhang Theory

External debt might affect economic growth through the debt overhang effect; this is the case when debt servicing discourages current as well as future investment plans (Ejigayehu 2013). According to Savvides (1992), a debtor country's failure to pay its foreign

debt might be connected to the country's economic situation. This type of country benefits little from increased productivity or export profits since a portion of the money is spent to pay off future debt. In this way, the debt overhang can be treated like a marginal tax rate on the country, which lowers return on investment and a hindrance to domestic capital formation.

Even in the condition all external debts are owned by government, debt overhang has a negative effect on private saving and investment (Ejigayehu 2013). "Debt overhang" is seen as a major source of economic distortion and slowing down of economic growth. This theory is applicable is this study because China has been blamed for saddling Africa with debt (Maeko 2021; Stones and Chazan 2020; Stein and Uddhammar 2021; White 2021). This could affect Africa's economic growth through the debt overhang effect.

### 2.1.4. Credit Rationing Effect

According to Borensztein, foreign debt affects economic growth through the credit rationing effect. This is a condition faced by countries that failed to obtain a new loan because of their inability or willingness to pay. African countries are already facing this challenge. China has been worried about Africa's indebtedness (Munda 2021; Pilling and Hille 2021). This has made it reduce its loan offerings to African countries (Du Plesis 2021; Wei 2021). Olander (2020) stated that, due to rising debt sustainability concerns, Chinese infrastructure funding to African nations is projected to decline in the coming years.

### 2.2. Empirical Literature

The literature has shown that infrastructural loans have a profound effect on growth and development. African is using China's loans to develop its infrastructure. This could have positive effects on economic growth. This section investigates the literature that has explored the contribution of infrastructure on development and also explores literature that looked at how foreign loans can affect economic growth.

Calderón and Servén (2003) used GMM estimates of a Cobb–Douglas production technology for a panel of 101 countries for the period of 1960–1997 and found positive and significant output contributions of three types of infrastructure assets: telecommunications, transport, and power for Latin American countries. Fedderke et al. (2006) used the endogenous growth theory to show that investment in infrastructure leads to economic growth in South Africa directly and indirectly.

Olukayode (2009) found a long-run association between public expenditure on infrastructure and economic growth over the period from 1977 to 2006 for the case of Nigeria and the Economic Community of West African States (ECOWAS) countries. Goetz (2011) concluded that transport infrastructure contributes to economic development by granting easy access to resources, lowering the costs of intermediate purchases and inducing higher productivity in manufacturing and production.

Sahoo and Kumar (2012) examined the output elasticity of infrastructure for four South Asian countries viz., India, Pakistan, Bangladesh, and Sri Lanka using panel cointegration techniques for the period of 1980–2005. The study found a long-run equilibrium relationship between output and infrastructure along with other relevant variables, such as gross domestic capital formation (GDCF), labour force, international trade, and human capital. Button and Yuan (2013) conducted the Granger causality test for panel data, which covered 35 airports and 32 metropolitan statistical areas in the USA over the period from 1990 to 2009. Their result indicated that air transport is a positive accelerator for domestic economic growth.

Kodongo and Ojah (2016) used a System GMM to estimate a model of economic growth augmented by an infrastructure variable, for a panel of 45 Sub-Saharan African countries, over the period of 2000–2011. They found that it was the spending on infrastructure and increments in the access to infrastructure that influenced economic growth and development in Sub-Saharan Africa.

Owusu-Manu et al. (2019) used an autoregressive distributed lag (ARDL) framework to determine the long- and short-run impacts of the selected infrastructure stock and quality indices on Ghana's economic growth. The findings indicate a statistically significant relationship between infrastructure development and economic growth. Rehman (2019), using the Autoregressive Distributed Lag (ARDL) bounds testing approach to cointegration, revealed that the level of electricity access available to the urban population contributed significantly to economic growth in Pakistan over the period from 1990 to 2016.

Wang et al. (2020) investigated the impact of transport infrastructure (railway and road) on the economic growth in the BRI countries. The estimation results at the national level reveal that the transport infrastructure in the BRI countries plays an essential role in facilitating economic growth. The positive spatial spillover effect of transport infrastructure on economic growth is most pronounced in Central and Eastern Europe.

This indicates the polarization effect in the initial stage of the lagging transport infrastructure and the diffusion effects after the transport infrastructure is mature. Chin et al. (2021) examined the role of infrastructure on economic growth in belt and road participating countries. The empirical results signify the importance of infrastructure development on economic growth in both the long-run and short-run. In addition, it is evident that capital and expenditure on health and education as well as exports, will accelerate economic growth.

Studies that investigate the relationship between physical infrastructure and economic growth mostly conclude that there exists a positive relationship. Some studies, such as Lall (1999); Roy et al. (2014), and Shi et al. (2017), however, found either no relationship or a negative relationship between infrastructure investment and economic growth. The effect of China's loans on Africa's growth could either be positive or negative. Although much of the literature suggested that infrastructure can promote growth, the manner with which the infrastructure funding (loans from China) process has been seen as unproductive.

It is thus important to examine the effect of foreign loans on economic growth. China has been saddling Africa with debt, and this could cause problems in the future. In countries with heavy indebtedness "debt overhang" is considered a leading cause of distortion and slowing down of economic growth (Sachs 1989). Economic growth slows down because these countries lose their pull on private investors. Additionally, the servicing of debts exhausts much of the indebted country's revenue to the extent that the potential of returning to growth paths is abridged (Levy-Livermore and Chowdhury 1998).

The IMF (2012) argued that, if borrowed public funds are used in productive activities, movements in future domestic interest rates, taxation rates, and debt service payments will not be injurious to the economy. (Clements et al. 2005) noted that, at low levels of debt, additional foreign borrowing could stimulate growth, to the extent that the additional capital financed by this new borrowing enhances the country's productive capacity.

Hansen (2001) found no important adverse association amongst external debt and economics growth, hence, the total absence of debt overhang. This was in a case of 54 underdeveloped nations. Edo (2002) examined the determinants of foreign debt accumulation with specific attention on Nigeria and Morocco. The study deduced that foreign loans servicing and accumulation has seriously and negatively impacted on the two countries and have severely and adversely affected investment.

Wijeweera et al. (2005) asserted that nations that suffer from debt overhang were those economies that found themselves on the wrong or bad side of the Laffer curve due to high debt accumulation resulting in debtors' inability to service the debts. Hameed et al. (2008) confirmed that the debt maintenance costs had adverse effects on the resultant output of principal and labour, which ultimately led to a decline in economic growth in Pakistan. Dey and Tareque (2020) examined the impact of external debt on economic growth in Bangladesh within a broader macroeconomic scenario. Their study revealed negative impact of external debt on GDP growth.

Ndoricimpa (2020) re-examine the threshold effects of public debt on economic growth in Africa. Low debt was found to be growth neutral, but higher public debt was growth

detrimental. Chindengwike and Kira (2021) examined the impact of foreign debts on economic growth in Tanzania. The study concluded that he long-term external debt stock did not have a significant effect on economic growth. In contrast, a short-term external debt appeared to have a significant impact on economic growth.

Olamide and Maredza (2021) examined the causal relationship between corruption, economic growth and external debt. The study showed that external debt servicing exert negative influences on economic growth while the effect of investment on growth was positive. Makhoba et al. (2021) examined the dynamic asymmetric relationships between public debt and economic growth in Southern African Developing Communities (SADC), over the period of 2000–2018. The low-debt regime was found to be positive and statistically significant, while the high-debt regime was detrimental for long-term growth.

Nermin (2021) investigated the relationship between external borrowing and economic growth in the Commonwealth Independent States during the period of 1995–2018. The obtained results suggest that there was a negative long-term unidirectional causal relationship running from external debt to GDP, thus, presenting strong evidence of the existence of the debt overhang hypothesis.

The next section focuses on exploring the dynamics of China's loans. Due to the novelty of this topic, the study had to consult both scholarly literature and newspaper articles. The sourcing of information from newspaper articles was guided by the inclusion and exclusion criteria. Establishing inclusion and exclusion criteria is a standard, required practice when designing high-quality research protocols. Inclusion criteria are the elements of an article that must be present in order for it to be eligible for inclusion in a literature review (Patino and Ferreira 2018).

In this study, only well-known and reputable news articles that reported a China–Africa story and quoted government officials were included. On the other hand, the exclusion criteria are the elements of an article that disqualify the study from inclusion in a literature review. In this study, articles that did not quote government officials were excluded (Patino and Ferreira 2018). This was done to ensure that the information obtained from these newspaper articles was credible and trustworthy.

## 3. The Dynamics of the China–Africa Loans

China's role as a source of loans in Africa is one of the least understood parts of China's partnership with Africa. Several analysts, newspaper articles, and scholars have raised different and competing views about the nature and dynamics of China's engagement with Africa. This section presents the competing views about China's development finance in Africa.

### 3.1. Need for Infrastructure Funding in Africa

Infrastructure development is an important means to promote development and growth. Perkins et al. (2005) argued that the "relationship between an economy and its economic infrastructure is analogous to that between a building and its foundation". Infrastructure supports a country's economic potential, and there is no country that can thrive economically and socially without a solid infrastructure base. Edinger and Labuschagne (2019) noted that infrastructure also increases business confidence and draws in investments in other sectors, fosters innovation and productivity, and lowers transaction costs, facilitating trade in goods and services and the transfer of talent.

It can thus be said that investment on infrastructure promotes economic activities and lack of infrastructure hampers sustainable economic growth and development initiatives (Sahoo et al. 2010). The costs of having poor infrastructure are not just the opportunity costs of lost growth. Poor infrastructure also affects human development. Several health problems, such as high child mortality, are caused by low access to basic services, such as electricity and clean water (African Development Bank (AfDB) 2019). This shows that "infrastructure is a *sinequanon* of growth and economic development and its development is not a luxury but a necessity" (Ranade 2009).

The provision and maintenance of adequate infrastructure facilities is essential for achieving sustainable economic development. One of the reasons that has led to the poor performance in developing countries' industries is lack of infrastructure (UN 2015; Chimbelu 2019). Without adequate infrastructure, the economy stagnates and the government itself will underperform in providing services. For instance, unreliable and expensive cost of electricity continues to inhibit the development of larger manufacturing sectors, while poor road and transportation infrastructure increases the cost of trading across borders, weighing on the overall competitiveness.

This has been the case in Africa; the African continent has a huge infrastructure gap (Campos 2018; Chimbelu 2019). This huge infrastructure deficit remains a hindrance to growth, investment and economic diversification. Inadequate infrastructure could be the main hindrance to Africa's development. About 60% of the continent's population does not have access to modern infrastructure, which isolates communities, prevents access to socio-economic facilities, and retards economic development (UN 2015; Campos 2018).

Addressing Africa's infrastructure deficit remains key to facilitating economic growth and stimulating productivity for Africa. Gondo (2019) stated that the World Bank claims that reducing Africa's infrastructure quantity and quality deficit has the potential to increase GDP per capita by as much as 2.6% per annum. Africa's minimum infrastructure needs stand at US$ 170 billion per annum (African Development Bank (AfDB) 2019). More than half of that requirement is presently unfunded (Edinger and Labuschagne 2019). Many African governments do not have the means to fund their basic services, such as education and health let alone infrastructure.

This may suggest that the challenge of constructing infrastructure cannot be met only by African governments because they lack adequate local revenue bases (Dollar 2019; BDlive 2018). This has resulted in many of Africa's infrastructure projects being funded by external forces. For instance, over the last couple of decades, China has assisted in meeting some of Africa's infrastructure financing needs and is now the single largest financier of African infrastructure (Soule 2019).

China has propelled from being a moderately minor lender in the continent to becoming Africa's main lender offering development finance that has assisted the continent record enormous expansion of basic infrastructure in the transport, water, and energy sectors. Chinese support for infrastructure development in Africa is related to its own local experience: infrastructure developments are vital to economic growth (Gordon 2019).

### 3.2. An Overview of Chinese Loans to Africa Since 2000

Chinese loans have become an important source of infrastructure finance for African countries over the past two decades. Figure 1 provides an overview of the trend and quantity of Chinese lending in Africa since 2000. Figure 1 shows that China's loans to Africa have been rising since 2000. Between 2000 and 2019, Chinese financiers committed around $153 billion to African governments and state-owned firms (Acker 2021; Wei 2021).

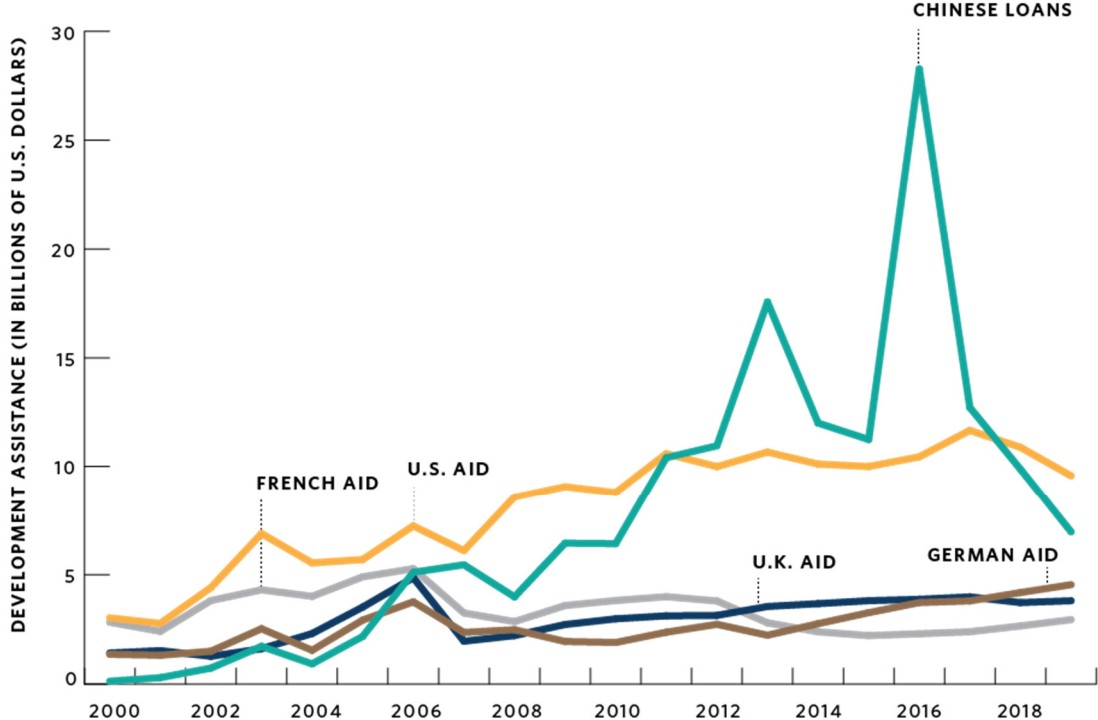

**Figure 1.** China's lending to Africa in comparison with traditional donors. Source: Adapted with permission from Usman (2021). Copyright 2021 Carnegie Endowment for International Peace.

It is also important to note that, around 2011, China replaced traditional Western lenders as the region's largest creditor. China's assistance to African countries accounted for 44.65% of its total assistance, which is substantially ahead of other regions in the world (Wei 2021). This move was influenced by African governments' emphasis on infrastructure development as well as China's readiness to lend to the region (Were 2018). China's loans are mostly used to fund infrastructure projects, and traditional lenders (Western nations and institutions), on the other hand, focus more than half (55%) of their financial assistance—a mix of grants and loans—on social sectors, like health, population, education, and humanitarian aid.

Unlike other major economies, much of China's external lending is official, meaning that it is undertaken by the Chinese government, state-owned policy banks, or state-owned commercial enterprises and banks (Horn et al. 2021). As can be seen from Figure 1, the flow of Chinese loans to Africa fell drastically in 2016. This has been seen as a sign of Beijing's growing caution over the continent's indebtedness (Munda 2021; Pilling and Hille 2021). Nyabiage (2021) concurred and stated that China's lending to Africa dropped by nearly 30 percent in 2019 after Chinese financiers stopped reduced the flow of loans into African countries with debt distress. China, however, remained the top bilateral lender to African countries in 2021 (Horn et al. 2021; Nyabiage 2021; Prentice and Strohecker 2021; Usman 2021).

### 3.3. African Governments View about Their Relationship with China

Many African people see grounds for positivity with regards to China's involvement in Africa. In an Afrobarometer survey of 36 African countries, almost two-thirds of respondents said China's influence was positive in their country, while only 15 % saw it as negative (AfroBarometer 2016). China's infrastructure and development finance initiatives are believed to be the reasons for China's positive image in Africa (Bombrowski 2017).

Africa's engagement with China has presented noteworthy benefits, mainly in reducing the continent's infrastructure gap (Kwasi 2019). Several African leaders have welcomed the economic partnership with China, which is seen as a contented substitute to the conti-

nent's long-standing Western engagements that regularly came with demanding conditions (Kwasi 2019). Chinese foreign ministry spokeswoman Hua Chunying argued that "the African people have the best say about China–Africa cooperation" (African Times 2018). Table 1 below shows some of the sentiments shared by African statesmen with regards to their relations with the Chinese.

**Table 1.** Selected African statesmen's views on their relations with the Chinese.

| |
|---|
| President Nyusi of Mozambique said that it is those who keep lecturing on the debt issue, not China, that caused the debt problem in Africa. |
| President Guelleh of Djibouti said that those criticizing the BRI have neither plans nor actions. Our confidence in the (Belt and Road Initiative) and trust in China and Africa-China cooperation will not waiver because of the groundless accusations. |
| Liberia's Economy Minister Augustus Flomo was quoted as saying: "China is a very, very important partner for our development strategy". |
| Namibian President Hage Geingob warned the US: "It assumes (US) Africans are children still," Geingob said of the oft-repeated concerns in the West over China's influence on the African continent. "They are not matured; they cannot deal with the world powers; they will be ill-treated and fooled as small kids. That's what it implies". |
| Uganda President Yoweri Museveni this week strongly defended his government's decision to take large Chinese loans, pushing back against his critics by claiming that his government really does not have any other choice but to turn to China. |
| Rwanda's President, Kagame's argued that Europe has neglected Africa, with investments that flow back to Europe but "leave nothing on the ground in Africa." "Instead of helping Africa it further impoverished the continent," Kagame said. |
| South Africa's President, Ramaphosa claimed that some people displayed a brazen prejudice against China as a financial partner for development, adding that if a Western organisation offered the same loan it would not be viewed with such scepticism. |
| Kenya's President Uhuru Kenyatta remarked in a pre-summit interview with *CGTN* that, "I believe it has huge potential for a win–win situation for all who are involved, and that is why we are very keen as a country, and I believe also as a continent, to partner strongly with China". |
| Ghanaian President Nana Akufo-Addo said his country wants to duplicate China's success story. The President further stated that he was "inspired by this model, and are trying to replicate same". |
| Zimbabwe's President, Emmerson Mnangagwa hit out at critics of China in an interview on state TV. The Zimbabwean President was quoted as saying; "There is now a transition to a new world order and those who don't see it are blind". |
| Djibouti's Finance Minister Ilyas Dawaleh: "Djibouti's development needs all its friends and strategic partners. At the same time, no one can dictate to us who we should deal with." |

Source: (African Times 2018; Cascais 2018; Dahir 2019; Withnall 2018; Bland 2018; Olander 2019).

Table 1 shows that China's activities in Africa are supported by African leaders themselves. This may support the claim that supports of China's economic activities in Africa argue that the scope, magnitude, and manner of Chinese aid practices are poorly understood and frequently misrepresented in the media (Hanauer and Morris 2014). It is also further argued that most countries have a constricted understanding of what China is able to provide (International Relations and Co-operation Minister Lindiwe Sisulu cited in Letsoalo et al. 2018). The sentiments from Table 1 may suggest that the Chinese African relations are based on mutual trust and respect. This may be the reason why some Africa countries have been given reprieve by China. This may also suggests talk of Africa's heavy indebtedness to China has been somewhat overblown (Olander quoted in Kazeem 2020; Bräutigam 2019).

### 3.4. Why Chinese Loans Are Preferred

Despite the presence of traditional funders, such as the World Bank, IMF, multilateral agencies, private creditors, and other Western nations, most African states see China as

their best bet for financing (Bavier and Shepherd 2018). Chinese loans are attractive from the viewpoint of African governments. There is a need to analyse why Chinese loans are attractive to African states. Analysing why China loans are attractive helps one to understand the complications tied to loans and development finance into Africa. Table 2 shows the pros and cons associated with borrowing options in Africa.

**Table 2.** Pros and Cons associated with borrowing options in Africa.

| Financing Source | Advantages | Disadvantage |
|---|---|---|
| China | Low interest Long maturity and grace period National sovereignty Policy autonomy | Lack of transparency Closed financial loop |
| IFIs (IMF and World Bank) and the rest of the world | Interest rates depends on the country rating Financing or support targets inclusive economic growth | Conditions, such as quantitative performance Conditions increasingly touch sensitive economic policies |
| International Market (Eurobonds) | Less restrictive and conditions Huge sums are available | High costs Short or medium term maturity |

Source: Adapted with permission from Afrodad (2019). Copyright 2019 Carnegie Africa Portal.

Table 2 shows that the Chinese loans do not come with so many conditions and their interest rates are "fairly comparable" to those imposed by other multilateral and private institutions. Furthermore, the loans have longer grace period when compared to borrowing from the IFIs, the rest of the world and the international market. It must, however, be noted that an empirical comparison of China's loans and those of other international lenders is difficult because China's loan contracts contain "unusual" confidentiality clauses, which prohibit borrowers from disclosing terms and conditions of the engagement (Chaudhury 2021; Peterson Institute for International Economics 2021).

Despite this, African countries continue to choose Chinese financing. Speaking at the Investing in Africa's Future conference in 2020, Senegalese President Macky Sall stated that loans from China are attractive to African countries due to the long-term financing with low-interest rates that they cannot find elsewhere (Nyabiage 2020). Beijing offers far better terms than Western banks, and European nations and the United States fail to match its generosity (Bavier and Shepherd 2018).

Table 2 also shows that China respects a nation's sovereignty when offering loans. China's has a hands-off approach, which emphasizes that only Africa can solve its political problems and does not make this resolution a prerequisite for development assistance. When the Chinese government interacts with African governments it does not present itself as the solution to and expert on Africa's problems (Were 2018). This is contrary to what Western nations and institutions do when offering aid and loans. Rwandan President, Paul Kagame, once argued that the West has an attitude of "adult supervision" and that they reduce Africa to the lowest common denominator (Bizimungu 2020).

China's assistance is often considered more attractive by recipient countries because it has few or no political strings attached and is often disbursed much more quickly and efficiently than assistance from Western nations. Consequently, while traditional donors have criticized China's approach to aid, many African countries embrace the assistance from Beijing or, at least, are glad to have more options (Li 2017). China's investment in Africa does not come with any political conditions attached and will neither interfere in internal politics nor make demands that people feel are difficult to fulfil (China's President, Mr Xi, quoted in Fifield 2018).

### 3.5. Criticism of China's Infrastructure Funding in Africa

Africa's traditional partners, particularly the United States and Europe, have questioned the reasons and motive of Chinese loans in Africa on several occasions (Moore 2021). China's activities in Africa have been questioned by, mainly, Western nations who believe that China impedes Africa's growth.

#### 3.5.1. Vanity Projects

China has been criticized for funding vanity infrastructural projects in Africa. Some projects have been seen to be white elephants, while others have failed to generate enough return to service the original loans used to fund them. Some analysts note that China uses development financing to support extremely visible schemes and projects, such as cultural centres, stadiums, and government facilities that provide limited or temporary economic benefits (Will 2012; Strange et al. 2013). Bräutigam (2018) noted that some of the projects financed and constructed by China are no doubt pork barrel projects and white elephants: airports with few passengers or bridges to nowhere.

In 2018, Sierra Leone cancelled a \$400 m (£304 m) Chinese-funded project to build a new airport outside the capital Freetown. Aviation Minister Kabineh Kallon told the BBC that a Chinese project, which was due to have been completed in 2022, was not necessary (BBC 2018). Tanzania also made a similar stance; it put on hold a project that was being operated by the Chinese. Tanzania's Bagamoyo Port was set to be the home of the continent's largest port—built and operated by the Chinese. A newly elected government did not agree with the terms of the \$10 billion project the country's previous administration made with Beijing. Tanzanian officials claimed that the deal was not commercially viable (Chimbelu 2019).

China has on several instances denied that it funds vanity projects. China defends its continued lending to Africa on the grounds that Africa still needs debt-funded infrastructure development (BDlive 2018). China's president, Xi Jinping was quoted as saying "China's cooperation with Africa is clearly targeted at the major bottlenecks to development. Resources for our cooperation are not to be spent on any vanity projects but in places where they count the most" (Chinese president Xi Jinping quoted in BDlive 2018 and in Goldsmith 2019). Furthermore, China maintains that negotiations and contracts are reached through equal consultations with no economic or political conditions.

China argues that African countries are making choices between Western offers and Chinese offers. Many developing countries prefer to use Western finance for things like budget support, health, and education, while turning to Chinese finance for big projects in transport and power (Dollar cited in Faga 2019). For instance, in 2019, Ethiopia's Prime Minister Abiy Ahmed cancelled the contract of a state-run military conglomerate to build the dam's turbines.

The Ethiopian Prime Minister said, at the time, that not a single turbine was functioning more than seven years after the government awarded the contract to the state-run military conglomerate. The contract was then offered to a Chinese company, China Gezhouba Group Co., Ltd. (Embassy of Ethiopia in Brussels 2019). This may suggest that China constructs projects that have that provide boundless economic benefits. It has also been argued that these loans are being used for developmental purposes.

Bräutigam (2018) found that China had lent at least \$95.5 billion between 2000 and 2015 and these Chinese loans were performing a useful service: financing Africa's serious infrastructure gap. However, the People's Daily, the Chinese Communist Party's mouthpiece, warned that Beijing should "pay more attention to how projects connect with the development and basic interests of relevant countries (Feng and Pilling 2019). China seemed to have heeded to this call. In 2018, China's president, Xi Jinping, said that "vanity projects" must be avoided in favour of more sensibly conceived initiatives that address confirmed economic bottlenecks (Feng and Pilling 2019).

### 3.5.2. Predatory Loans

In 2018, former United States secretary of state Rex Tillerson blamed China of providing "predatory loans", which weaken growth prospects and job creation on the African continent (Bräutigam 2018). These "predatory loans" have also been seen to be contributing to the increase in debt levels in Africa. In August, 16 American senators voiced their concern about "predatory Chinese infrastructure lending" (BBC 2018).

It is claimed that Africa countries are burdening themselves in substantial amounts of infrastructure-induced debt that they may not be able to afford. Already there are some cautionary signs: the Addis Ababa-Djibouti railway line is believed to have cost Ethiopia approximately 25% of its 2016 budget, and Kenya's railway from Nairobi to Mombasa, which was financed by the Chinese, went four times over budget (Wade 2019). In Djibouti, by the end of 2016, 82% of external debt was owed to China (Kwasi 2019).

Were (2018) argued that the argument that China plies naïve African states with debt fails to recognize that African states are aware of these debt commitments. Furthermore, there are some who claim that the West also have contributed to the debt that Africa has and only a portion of Africa's overall debt is actually owned by China. Bavier and Shepherd (2018), and Eysse and Welle (2018) indicated that the debt that China has contributed in Africa in not yet as high as compared with Western Nations, such as the US, the UK, and France.

Dahir (2019) concurred and stated that Chinese loans are presently not a main contributor to the debt burden in Africa; much of the debt in Africa is still owed to institutions like the World Bank and the debt owed to the institutions is more expensive to service than Chinese loans. Anthony cited in Donnelly (2018) concluded that the debt-trap narrative fits into a Euro-American discourse but surprisingly this discussion is frequently left out of the debt-trap discourse. The Chinese, themselves, deny that they are responsible for Africa's financial debt. For instance, during his trip to Ethiopia, Chinese Foreign Minister Wang Yi emphasized that "we know about some African nations' financing difficulties," adding that his country had nothing to do with them, though (Rostek-Buetti 2019).

### 3.5.3. Control and Dominance

There is a belief that China is funding projects in Africa in order to assert control and dominate African countries in the future. China's main aim with foreign investment is geostrategic, not commercial (Mourdoukoutas 2019). There is a belief that China's own commercial and geopolitical interests are maximized when its lending allies are suffering. Chinese policy had the potential of affecting port calls and hub status decisions (Hutson 2018).

Chinese investments in sub-Saharan African ports have been seen to be allegedly posing threats to U.S. influence in sub-Saharan Africa as well as African independence. A CSIS report, *Influence and Infrastructure: The Strategic Stakes of Foreign Projects*, identifies some of the strategic risks posed by the Chinese infrastructure projects; permitting Beijing to possibly limit access to its rivals, exploit ports during conflict, and gather intelligence information (Devermont and Chiang 2019).

There are potential threats, such as the taking over of African ports by the Chinese to the detriment of Africans (Hutson 2018). Dahir (2019), Bavier (2019) and Pandey (2019) cite Sri Lanka's Hambantota Port as a warning account of the dangers of dependence on Chinese loans. It was alleged that Chinese state-owned company took control of the Sri Lankan port of Hambantotaat the southern tip of India.

However, some investigations have shown that severe economic factors forced the government to seek out for various ways to raise foreign currency and leasing Hambantota port to a Chinese investor, which was not generating sufficient return on investment (Moramudall 2019; Jones and Hameiri 2020). This may suggest that Sri Lanka was not in any way coerced to surrender its port to China. Bräutigam (2020) conducted a study and found no evidence of asset seizures in Africa, or indeed, anywhere among Chinese borrowers in debt difficulties.

It is for this reason that Park and Benabdallah (2021) argued that the U.S. strategy to "counter" China in Africa (and elsewhere) has resulted in generalizations about China–Africa relations that depict Chinese officials and companies as predators and Africans as their hapless victims. However, certain African governments seem to be aware of China's control. In 2018, the South African parliament called for a disclosure of a Chinese loan to Eskom, a power utility and to Transnet, a transport utility. President Cyril Ramaphosa reassured members of Parliament that Eskom would not be taken over if it defaulted on its $2.5bn (R33.4bn at the time) loan from the Chinese Development Bank (Phakathi 2018).

3.5.4. Debt Trap Diplomacy

There is a belief that China is saddling Africa with debt. Mihalyi cited in Zimbabwe Coalition of Debt and Development (2021) argued that, even though these loans have often provided much-needed infrastructure, such as roads and hydro-dams, in several cases, they have led to increasing and unsustainable levels of debt, and the danger of losing the collateral is worth more than the value of the loan. Some argue that China is saddling Africa with unsustainable debt and seeks to use indebtedness to further its geopolitical control over the continent.

For instance, Su (2017) argued that, by ensuring that 'debts are paid in some form or the other, whether it is economic concessions, political agreements, or a combination of both, China may in the long term formulate a new kind of diplomatic relationships with Africa. The magnitude of China's lending to African countries as well as China's motives has been labelled as "debt trap diplomacy". China's debt trap diplomacy is often labelled as a long-term aim to trap African countries in debt commitments they cannot fulfil and eventually leverage that pretext to obtain African resources or important assets (Kazeem 2020).

**4. Methodology**

*4.1. Data Sources and Research Approach*

A small number of studies have examined the impact of Chinese aid and loans cross-nationally due to the scarcity of reliable data, relying instead on case studies (Chen and Kinzelbach 2015; Hackenesch 2015; Li 2017). This study avoids this challenge by using secondary data from the China–Africa Research Initiative at Johns Hopkins University. In order to test the impact of China's infrastructural loans on economic growth in selected African states, a quantitative approach was used.

The decision to use a quantitative analysis was based on the fact that a literature survey showed that the reports on China–Africa relations are subjective and somewhat speculative. They are also driven by an agenda that is clearly not about reflecting the situation on the ground. A quantitative approach was, therefore, deemed appropriate in separating facts from speculation.

The study used secondary quantitative panel data. The data used in this study are obtained mainly from the World Bank as well as the China–Africa Research Initiative at Johns Hopkins University. The study employed annual panel data covering the period of 2000–2018. The study used 15 African countries that are debt distressed. A country is seen as being in debt distress when it is struggling to service its debt, as demonstrated by arrears, the restructuring of its debt or other clear signs that a debt crisis is looming (Mustapha and Prizzon 2018; UN 2021).

Chad, Congo Republic, Eritrea, Mozambique, South Sudan and Zimbabwe were considered to be in debt distress at the end of 2017 while Zambia and Ethiopia were downgraded to "high risk of debt distress" (Madowo 2018). Angola, the Democratic Republic of the Congo (DRC), Djibouti, Ethiopia, Kenya, Mozambique, Sudan, and Zambia are some of the countries that are highly indebted to China and highly debt distressed (Kwasi 2019). This was the primary motivation for choosing debt-distressed countries.

Africa's rising debt to China is part of a broader concern over the continent's debt sustainability. China's Silk Road project, from which African states have gained immensely

by obtaining commercial loans for infrastructure development, has returned to haunt some African countries who are now facing debt distress (Fabricius 2021; Olingo 2021).

Unsustainable debt is undesirable for African countries. It is argued that high public debt can negatively affect capital stock accumulation and economic growth via heightened long-term interest rates, higher distortionary tax rates, inflation, and a general constraint on countercyclical fiscal policies, which may lead to increased volatility and lower growth rates (Calderón and Fuentes 2013; De Rugy and Salmon 2013).

### 4.2. Model Specification

The empirical model of this study is related to the work of Chin et al. (2021). Chin et al. (2021) investigated the role of infrastructure on economic growth in belt and road participating countries. The study chose the Chin et al. (2021) model because their study attempted to answer the same question that this study seeks to answer. Their study included African countries and other developing countries in its study sample. This study also focusses on African countries and the countries included in this study share common economic characteristics that make them similar enough for the experiences to be compared. This study builds on Chin et al. (2021) and formulates the following model:

$$GDP_{it} = \beta_0 + \beta_1 RR_{it} + \beta_2 L_{it} + \beta_3 IMP + \beta_4 FDI_{it} + \beta_5 AGR_{it} + \varepsilon_{it} \tag{1}$$

where *GDP* is the Gross Domestic Product is, *RR* is Resource Rents, *L* is Chinese Loans to African Governments, *IMP* is imports, *FDI* is Foreign Direct Investment, *AGR* is Agricultural growth, and $\varepsilon_{it}$ is an error term. The description of the variable presented in Equation (1) above is presented in Table 3 below.

**Table 3.** Summary of variable descriptions.

| Variable | Description and Unit of Measurement | Source |
|:---:|:---|:---:|
| RR | Total natural resources rents are the sum of oil rents, natural gas rents, coal rents (hard and soft), mineral rents, and forest rents. | World Bank |
| L | Chinese Loans to African Governments. These loans are offered mainly by Exim Bank of China, China Development Bank. These provide approximately 80% of the loans, and they are also controlled by the Chinese government. The rest is provided by other small state controlled banks and private banks. | China–Africa Research Initiative |
| IMP | Total imports of goods and services. | World Bank |
| GDP | This is the Annual percentage growth rate of GDP (in PPP standards) at market prices based on constant local currency. | World Bank |
| FDI | Foreign direct investment refers to direct investment equity flows in the reporting economy. | World Bank |
| AGR | Agriculture, forestry, and fishing, value added (% of GDP) | World Bank |

The choice of variables was guided by literature. Imports were chosen because Africa imports more than it exports (World Bank 2018). Furthermore, some of these imports, particularly those from China, have been seen to be causing deindustrialisation. This created a need to test their contribution to Africa's economic growth. Agriculture was chosen because the agricultural sector accounts for at least 23% of Africa's GDP with about 40% of the workforce engaged in the sector (Kufuor 2021). Resource rents were included because in 2018, total natural resource rents accounted for 16 percent of gross domestic product in African countries much higher than world average (World Bank 2018). This shows that it a significant sector in Africa.

### 4.3. Estimation Techniques

The study subjected its data to several pre-tests in order to determine the correct estimation technique. The preliminary tests that were done are the unit root tests and cointegration tests. After the presence of cointegration was found, the study proceeded to perform a panel ARDL approach.

#### 4.3.1. Unit Root Tests

Though testing for the order of integration of variables is not important when applying the ARDL model as long as the variables of interest are I(0) and I(1), (Pesaran et al. 1999), this study carried out these tests just to make sure that no series exceeds I(1) order of integration (Samargandi et al. 2013; Aliha et al. 2017). The study used the LM, Pesaran and Shin test to test for unit root. This test has been widely implemented in empirical research due to its rather simple methodology and alternative hypothesis of heterogeneity (Afonso and Rault 2007).

#### 4.3.2. Panel Cointegration Test

In order to check the cointegrated relationship among the variables, the study employed the Pedroni and Fisher cointegration tests. Pedroni (2004) provided cointegration tests for heterogeneous panels based on the two-step cointegration approach of Engle and Clive (1987).

Pedroni uses the residuals from the static (long-run) regression and constructs seven panel cointegration test statistics: four of them are based on pooling (within-dimension or 'panel statistics test'), which assumes homogeneity of the AR term, whilst the remaining are less restrictive (between-dimension or 'group statistics test') as they allow for heterogeneity of the AR term. The second cointegration test was the Fisher Test. Maddala and Wu (1999) proposed a Fisher cointegration test based on the multivariate framework of Johansen (1988). They suggested combining the *p*-values of individual (system-based) cointegration tests in order to obtain a panel test statistic.

#### 4.3.3. Panel ARDL

This study followed a panel ARDL approach after it was found that the variables had mixed levels of integration. The study chose to use the panel ARDL method because the variables had mixed orders of integration (Mushataq and Siddiqui 2017). Other panel cointegration techniques, such as the Dynamic Ordinary Least Squares (DOLS) and Fully Modified Ordinary Least Squares work better when the variables' order of integration are not mixed. The panel ARDL approach was also followed because it is very efficient even for mixed orders of integration and small sample sizes having 30 to 80 observations (Baig and Qayyum 2018). The study performed two panel ARDL approaches, which were the Mean Group estimator and the Pooled Mean group estimator.

#### Mean Group (MG) Estimator

The MG, introduced by Pesaran and Smith (1995), calls for estimating separate regressions for each country and calculating the coefficients as unweighted means of the estimated coefficients for the individual countries. The conventional estimation methods, such as the fixed effects, the random effects, and the GMM have the purpose of correcting the fixed-effect heterogeneity issue that occur in the case of large N and small T panels (Lin-Sea et al. 2019).

However, these estimators would produce inconsistent results as they do not take endogeneity caused by heterogeneity into consideration. To ensure consistent results, the mean group (MG) estimator introduced by Pesaran and Smith (1995) that tolerates differences in intercepts, slope, and error variances across groups can be used. This does not impose any restrictions. It allows for all coefficients to vary and be heterogeneous in the long-run and short-run (Manes et al. 2016).

Pooled Mean Group (PMG) Model

Pesaran et al. (1999) proposed a PMG estimator that combines both pooling and averaging. The main characteristic of PMG is that it allows short-run coefficients, including the intercepts, the speed of adjustment to the long-run equilibrium values, and error variances to be heterogeneous country by country, while the long-run slope coefficients are restricted to be homogeneous across countries. The MG estimator does not take the issue of cross-sectional dependence into account. Alternatively, the pooled mean group (PMG) estimator developed by Pesaran et al. (1999), which is more efficient due to the valid long-run restrictions, can be considered. Another advantage of the PMG over MG is that it is robust to the outliers and lag orders. The model for the PMG estimator is presented as below:

$$y_{it} = \sum_{j=1}^{p} \gamma_{ij} \, y_{it-j} + \sum_{j=0}^{p} \beta_{ij} \, x_{it-j} + \mu_i + \mu_{it} \tag{2}$$

$$\mu_{it} = \rho_i' \, f_t + \varepsilon_{it} \tag{3}$$

where $x_{it}$ represents $k \times 1$ vector of explanatory variables for group $i$, $\mu_i$ is the fixed effects, $\beta_{ij}$ denotes the coefficient vectors ($k \times 1$), $\gamma_{ij}$ refers to coefficients of the lagged dependent variables, and $f_t$ is a vector of unobserved common shocks (Lin-Sea et al. 2019).

Hausman Test

Lastly, there is a need to make comparison and choice between PMG and MG estimators in terms of efficiency and consistency. The Hausman test is commonly employed in this regard. The Hausman test is a test based on panel ARDL approach that measures the efficiency and consistency of the estimates of MG and PMG (Sulaiman 2020). This test was employed in this study to choose the most appropriate technique between the PMG and MG estimators.

## 5. Presentation of Results

### 5.1. Descriptive Statistics

Table 4 summarises the descriptive statistics of the variables that were used in this study.

**Table 4.** Descriptive statistics.

| Descriptive Stats. | L | GDP | FDI | RR | IMP | AGR |
|---|---|---|---|---|---|---|
| Mean | 713.9333 | 228.9263 | 6.02375 | 3.11239 | 35.85939 | 6.291874 |
| Median | 4235.5000 | 145.0000 | 1.59318 | 4.62888 | 31.02384 | 2.92456 |
| Maximum | 5933.000 | 1032.000 | 1.946273 | 11.119465 | 107.94636 | 5.39756 |
| Minimum | 50.000 | 26.4000 | 7.23950 | 2.395806 | 34.0298 | 379467 |
| Std. Dev. | 845.2909 | 339.8948 | 1.56734 | 3.90548 | 21.8570 | 8.12065 |
| Skewness | 3.89759 | 1.51295 | 1.830486 | 2.798452 | 0.772397 | 2.27295 |
| Kurtosis | 26.1254 | 3.69322 | 17.02645 | 11.223859 | 3.56749 | 10.28539 |
| Jarque–Bera | 4466.640 | 68.66179 | 45.90455 | 43.6965 | 43.6392 | 117.9344 |
| Probability | 0.00432 | 0.00000 | 0.00000 | 0.00000 | 0.00000 | 0.0000 |
| Sum | 74.148 | 39146 | 2.3.1930 | 8.10346 | 345.46 | 2.41532 |
| Observations | 270 | 270 | 270 | 270 | 270 | 270 |

The *L* variable was positively skewed and it had a value of 3.89 and the other variables had a values that were above 0. The *L* variable also had a high kurtosis value (26.125) and other variables had values that were higher than 3. Only *GDP* and *IMP* had low kurtosis values. Lastly, the Jarque–Bera statistic is below the 0.05% level of significance for all the series. This is an indication that the variables do not follow a normal distribution.

### 5.2. Stationarity Tests

The study started by checking the stationary properties of the data. The LM, Pesaran and Shin test was used and the results are reported in Table 5.

**Table 5.** Stationarity tests.

| Variable | LM, Pesaran and Shin Test | |
| :---: | :---: | :---: |
| | **Level** | **1st Diff** |
| *L* | −2.26718 | 5.2372 |
| *RR* | −1.3956 | 7.3531 |
| *IMP* | 3.9382 * | - |
| *FDI* | 1.2190 | 5.5126 |
| *AGR* | 4.2391 * | - |
| *GDP* | 1.35128 | 4.2901 |

* Denotes variable was stationary at levels.

Table 5 shows that, under LM test, IMP, and AGR were stationary at levels. All other variables had a unit root at levels. They became stationary at first differencing. This necessitated the use of cointegration techniques to examine whether or not there was a long run association amongst the variables. For the long-run and short-run estimates, the variables were estimated in their first difference

*5.3. Cointegration Tests*

The presence of cointegration was tested using the Pedroni test and the Fisher test. The results are shown in Tables 6 and 7.

The results from the Pedroni test shows that the null hypothesis of no cointegration cannot be rejected at the 5% significant level. This suggests that there is cointegration amongst the variables. The Fisher test was also performed and results are shown in Table 7 below.

**Table 6.** Pedroni Cointegration test.

| Test Statistics | Statistic | Prob. | Statistic | Prob |
| :---: | :---: | :---: | :---: | :---: |
| Panel v-statistic | 0.82398 | 0.8672 | −2.75298 | 0.0000 |
| Panel rho-statistic | −2.8516 | 0.0000 | −2.91102 | 0.0000 |
| Panel PP-statistic | 3.45762 | 0.0000 | −2.13921 | 0.0298 |
| Panel ADF-statistic | −0.981751 | 0.6349 | −2.27201 | 0.0348 |
| Group Panel rho-statistic | 2.93416 | 0.0198 | - | - |
| Group PP-statistic | −3.86283 | 0.0212 | - | - |
| Group ADF-statistic | −3.22713 | 0.0000 | - | - |

**Table 7.** Fisher panel cointegration test.

| Hypothesised No of CEs | Fisher Statistic (from Trace Test) | Prob. | Fisher Statistic (from Max-Eigen Test) | Prob. |
| :---: | :---: | :---: | :---: | :---: |
| None | 0.000 | 1.0000 | 0.000 | 1.000 |
| At most 1 | 197.4 | 0.0000 | 153.9 | 0.0241 |
| At most 2 | 43.2 | 0.2741 | 18.3 | 0.7321 |

The results show that both the maximal eigenvalue statistics and the trace statistics reject the null hypothesis of no cointegrating relationship. It can thus be said that the Fisher panel cointegration test support the hypothesis of a cointegrating relation. This confirms the Pedroni test results, which also found that there was cointegration amongst the variables under investigation. The next step was to apply a panel ARDL approach to scrutinize the long run relationship between China loans and economic growth in Africa.

*5.4. ARDL Results*

Since the pre-tests for unit-roots and cointegration suggest that the variables are non-stationary and cointegrated as assumed in Equation (1), the study proceeded to estimation

of the long-run and short-run relationships using the MG and PMG estimators. The Hausman test was also performed in order to make comparison and choice between PMG and MG estimators in terms of efficiency and consistency. The results are shown in Table 8.

**Table 8.** Hausman test (null hypothesis: there is long run homogeneity restriction).

| Chi-Square | *p*-Value |
|---|---|
| 7.92 | 0.913 |

The results show that the *p*-value is greater than 0.05, and this makes the study to fail to reject the null hypothesis at 5% significance level. This supports the suitability of the PMG model. The analysis will, therefore, rely on the PMG model for analysing the results. Tables 9 and 10 show the long-run and short-run results and the error correction results, respectively.

**Table 9.** Long-run panel ARDL results.

| PMG | | | | MG | | | |
|---|---|---|---|---|---|---|---|
| Variable | Coefficient | t-Statistic | Prob | Variable | Coefficient | t-Statistic | Prob |
| L | 2.140 | 2.2450 | 0.0263 | L | 2.333 | 2.4466 | 0.0155 |
| RR | −0.007 | −3.5756 | 0.0005 | RR | −0.0067 | −3.3521 | 0.0010 |
| LIMP | 2.344 | 7.2824 | 0.000 | IMP | 2.500 | 7.9096 | 0.0000 |
| LFDI | 0.0055 | 1.7351 | 0.0848 | FDI | 0.0047 | 1.5080 | 0.1336 |
| LAGR | −0.0047 | −1.2464 | 0.2146 | AGR | −0.0045 | −1.2069 | 0.2293 |

**Table 10.** Short-run results and ECM.

| PMG | | | | MG | | | |
|---|---|---|---|---|---|---|---|
| Variable | Coefficient | t-Statistic | Prob | Variable | Coefficient | t-Statistic | Prob |
| L | 0.3480 | 1.1059 | 0.2727 | L | 0.3206 | 1.1710 | 0.2458 |
| RR | 1.4216 | 1.1158 | 0.2684 | RR | 0.6569 | 1.5765 | 0.1196 |
| LIMP | 0.0363 | 4.7490 | 0.000 | IMP | 0.4424 | 20.5909 | 0.000 |
| LFDI | 2.0936 | 2.7305 | 0.0080 | FDI | 0.5773 | 12.959 | 0.0000 |
| LAGR | 0.4385 | 23.7795 | 0.0000 | AGR | 1.0368 | 0.8077 | 0.4221 |
| ECT | −0.3063 | −7.1117 | 0.0000 | ECT | −0.2058 | −7.1404 | 0.0000 |

Since the PMG was deemed superior than the MG estimator, its results were used and those of the MG estimator were discarded. Table 9 shows that the study found evidence of a positive relationship between Chinese loans and economic growth, which is in line with the existing literature; World Bank (2009), Kelly (2012) and Bräutigam (2018) and Gu and Carey (2019). The results seem to support the Chinese belief that that its economic relationships with African countries are mutually beneficial.

In one of its reports, the World Bank also recognized the substantial development opportunity that Chinese development finance presented to African countries (World Bank 2009). Kelly (2012) also evidence that suggested that there is a positive association between Chinese loans and African growth since 2009. Bräutigam (2018) also acknowledged the development opportunity offered by Chinese loans; on a continent where over 600 million Africans have no access to electricity, 40 percent of the Chinese loans paid for power generation and transmission.

The use of night-time satellite images to show that Chinese loans and projects (roads, digital systems and energy) done using these loans have positively contributed to growth and poverty reduction (Gu and Carey 2019). The China infrastructure schemes offer Africa with prospects for development and will possibly be significant driver of infrastructure

development in Africa (Eleanor 2017; Were 2018). Critics have argued that China's development finance to Africa may not produce any benefits. However, this study contradicted this and an increasing community of scholars is producing research pointing out the inaccuracies of this narrative (Park and Benabdallah 2021).

The study found evidence of a positive relationship between imports and growth in Africa. This suggests that imports have a positive impact on economic growth. The findings are inconsistent with Zahonogo (2016) who concluded that African countries must have more effective trade openness, particularly by productively controlling import levels, in order to boost their economic growth through international trade. However, the results are in line with Oyebanjo (2017). A study by Oyebanjo (2017) showed that that both exports and imports contribute significantly to economic growth in Africa. On a specific level, growth in raw material exports, and not manufactured exports, is significantly associated with GDP growth while growth in manufactured imports, and not raw material imports, is significantly associated with GDP growth.

A weak positive relationship was found between FDI and economic growth. Kargbo (2017) suggested that the impact of FDI on productivity growth differs across African countries. The study further showed that the effect of FDI was conditional upon other factors, such as human capital. Only countries that had a minimum threshold of 6.94 average years of schooling experienced productivity gains from FDI (Kargbo 2017). Sakyl and Egyir (2017) found that FDI inflows had a substantial effect on economic growth in Africa. The study found a weak negative relationship between RR and economic growth.

This is consistent with literature on the resource curse (Carmignani and Chowdhury (2010); Janda and Quarshie (2017); Ndjokou and Tsopmo (2017) and Zalle (2018)). Carmignani and Chowdhury (2010) argued that natural resource revenue negatively affects economic growth in Sub-Saharan Africa (SSA) because SSA specializes on primary commodities that are not conducive for growth. Zalle (2018) claimed that African countries must simultaneously strengthen investments in human capital and fight against corruption to turn the curse of natural resources into a blessing.

The short-run analysis presented in Table 10 shows that coefficient of the error correction term (ECT) is negative, and it has a coefficient that is lower than $-2$. This shows that the model used in this study met the requirements for the efficiency, consistency, and validity of a long run association among the variables under investigation (Samargandi et al. 2013). The coefficient of the ECT turned out to be $-0.3063$. This was significant at 1 percent.

This implies that the 31 percent of the disequilibrium is corrected each year. The short-run analysis also shows that loans do not have a significant impact on economic growth. This may suggest the benefits associated with Chinese loans are not realized in the short term. This makes sense because much of the loans are used to develop infrastructure, and it may take a while before the benefits of infrastructure projects are realized. Resource rents are also insignificant, and this suggests that, in the short run, resource rents do not affect economic growth.

The study performed additional panel cointegration techniques for robustness purposes. The Fully Modified Ordinary Least Squares (FMOLS) Dynamic Ordinary Least Squares (DOLS) were performed. The results are reported in Table 11 below.

**Table 11.** Long-run panel cointegration results.

| FMOLS | | | | DOLS | | | |
|---|---|---|---|---|---|---|---|
| **Variable** | **Coefficient** | **t-Statistic** | **Prob** | **Variable** | **Coefficient** | **t-Statistic** | **Prob** |
| *L* | 2.749 | 4.557 | 0.0000 | *L* | 0.870 | 4.3491 | 0.0000 |
| *RR* | 0.0078 | 0.0312 | 0.9751 | *RR* | 0.0566 | 0.5945 | 0.55534 |
| *LIMP* | 1.0783 | 6.2431 | 0.000 | *IMP* | 0.2605 | 2.2792 | 0.0245 |
| *LFDI* | 0.4542 | 1.9226 | 0.0570 | *FDI* | 0.0289 | $-0.5820$ | 0.5616 |
| *LAGR* | $-0.6550$ | $-3.9428$ | 0.0000 | *AGR* | 0.1242 | 5.1908 | 0.0000 |

The findings in Table 8 are upheld in the FMOLS and DOLS results in Table 10. The coefficient for China loans (L), has a positive sign, which implies that an increase in China infrastructural loans, could bring about an increase in economic growth in African economies. This finding is consistent with the recent literature. A study by (Ehizuelen 2021) showed that China's infrastructure investment has assisted the continent to deliver extraordinary headway in awakening African economies.

Marais and Labuschagne (2019) concurred and stated that the Belt and Road Initiative, benefiting from Chinese firms' access to preferential financing for overseas investments and exports, is making African economies more connected to one another and to the outside world, thereby, boosting economic diversification and growth. The other findings display that indicators, such as IMP and FDI, have a positive and significant influence in increasing growth. However, agriculture was seen to be contributing negatively to growth in the FMOLS model and contributing positively to growth in the DOLS model.

## 6. Conclusions

This study sought to test the impact of China's infrastructure investment on economic growth in Africa. A survey of the literature showed that the central question about Chinese infrastructural loans in Africa is whether they will be effective at establishing long-term sustainable development. A well-maintained infrastructure should be the base upon which the economy can flourish, and good infrastructure should provide opportunities for economic growth.

The provision and maintenance of adequate infrastructure facilities is essential for achieving sustainable economic growth. One of the reasons leading to the poor performance in African countries is lack of infrastructure. Although much of the literature suggested that infrastructure can promote growth, the manner of the infrastructure funding (loans from China) process has been seen as unproductive. Against this backdrop, the study sought to empirically tests the impact of China's infrastructure investment on economic growth in Africa.

The main conclusion from the quantitative analysis of China's infrastructural loans in Africa is that China's efforts in developing infrastructure are translating to economic growth. Given the evidence of this study, it is recommended that African governments should, with caution, embrace China's infrastructural loans. China has been and continues to be a major force in assisting African countries in financing their development. Based on the findings from this study, financial support from China should be welcomed in order to help Africa bridge its finance deficit.

However, it should be noted that, while China's infrastructure loans may lead to long-term economic progress, excessive amounts of foreign debt can be particularly dangerous for African countries. It is, therefore, critical for African governments to have policies in place to cope with debt obligations and to prevent a catastrophe due to debt overhang. A positive relationship between China loans and growth in Africa has been established; however, African governments should bear in mind that foreign borrowing only stimulates growth to the extent that the additional capital financed by this new borrowing enhances the country's productive capacity.

**Funding:** This research received no external funding.

**Institutional Review Board Statement:** Not applicable.

**Informed Consent Statement:** Not applicable.

**Data Availability Statement:** The study did not report any new data.

**Conflicts of Interest:** The author declares no conflict of interest.

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
