# Peer review of "China in Africa: An Examination of the Impact of China’s Loans on Growth in Selected African States"

_economies, doi:10.3390/economies10070154_

Round 1

Reviewer 1 Report

1.The author already chosen literature with a coherent story that is closed related to the purpose and research method of the paper.

2.It is suggested that the author may ask the MDPI editing center to revise the paper and submit it again.

Author Response

Thank you. We will consider sending it to the MDPI editing center. 

Reviewer 2 Report

My further comments are as follow:

  1. Figure 1 is very helpful. But can you also show how the fraction of Chinese loans changes over time? In addition, can you elaborate on the spike of Chinese loans around 2016? Do you have data on Chinese loans after 2018? If the trend continues, China would only be the No.1 lender to African countries for several years, this might threaten your argument about the importance of Chinese loans.
  2. In your data source section, can you provide summary statistics for your data?
  3. Can you put your response to my third point into your paper? It would be a bit odd to show the equation without explaining why these variables are included.
  4. Given that the US has been an important source of loans to African countries, can you run the same regression to show how the impact of Chinese loans on development is different from Chinese loans?

Author Response

Reviewer 2

  1. Figure 1 is very helpful. But can you also show how the fraction of Chinese loans changes over time? In addition, can you elaborate on the spike of Chinese loans around 2016? Do you have data on Chinese loans after 2018? If the trend continues, China would only be the No.1 lender to African countries for several years, this might threaten your argument about the importance of Chinese loans.

We added a brief explanation of the spike in Chinese loans around 2016. China reduced its loans to Africa because many African countries had failed to repay previous debts. This made China to scale down on its loans. However, literature argues that by 2021 China was still the biggest bilateral lender. Literature also states that China managed to arrange the debt repayment plans with Africa countries and it, then, increased its loans to Africa.

  1. In your data source section, can you provide summary statistics for your data?

The Summary of statistics have been added. Thank you so much for the comment; summary statistics are needed in every economic analysis.

  1. Can you put your response to my third point into your paper? It would be a bit odd to show the equation without explaining why these variables are included.

Thank you. We included this response under the data description table.

  1. Given that the US has been an important source of loans to African countries, can you run the same regression to show how the impact of Chinese loans on development is different from Chinese loans?

I am afraid to say that running a regression to compare the Chinese and US development assistance will make it seem like we were comparing the two states. Furthermore, it is common knowledge that the US’s development assistance has greatly assisted African countries. the reason why Western development assistance is not preferred these days is that many African governments do not want to account for their actions (corruption, human rights records, checks and balances etc).

Reviewer 3 Report

Review report on “CHINA IN AFRICA: AN EXAMINATION OF THE IMPACT OF CHINA’S LOANS ON GROWTH IN SELECTED AFRICAN STATES”
1. The definition of “infrastructure loans” in this paper should be clarified. Does it mean loans provided, subsidised, or guaranteed by the Chinese government? Or does it also include purely private sector loans? Additionally, it is useful to add information on, if possible, the providers and the characteristics of the loans such as conditionality, loans outstanding, destinations, etc.
2. L 106- 2.1.1 Financing Gap theory
Chenery and Strout (1966) highlighted that foreign aid could be used to fill either a savings gap or a foreign change gap needed for less developed countries' economic development. This model is called the “Two-gap model” and has been discussed extensively in the context of foreign aid effectiveness. Therefore, the discussion in this section could reflect this point.
3. L619- Table 3. Summary of Variable Description
Specify the unit of RR, L and IMP (in USD?). Check whether the variables used in the model are the constant price term.

Author Response

Review report on “CHINA IN AFRICA: AN EXAMINATION OF THE IMPACT OF CHINA’S LOANS ON GROWTH IN SELECTED AFRICAN STATES

  1. The definition of “infrastructure loans” in this paper should be clarified. Does it mean loans provided, subsidised, or guaranteed by the Chinese government? Or does it also include purely private sector loans? Additionally, it is useful to add information on, if possible, the providers and the characteristics of the loans such as conditionality, loans outstanding, destinations, etc.

The paper only considered loans that are given by China’s state owned enterprises. It is said that over 80% of the loans in Africa come from China’s state owned enterprises. This is why the Chinese government is actively involved in all the economic dealings with Africa. In other words, most loans and economic deals are done on a government to government arrangement.

  1. L 106- 2.1.1 Financing Gap theory
    Chenery and Strout (1966) highlighted that foreign aid could be used to fill either a savings gap or a foreign change gap needed for less developed countries' economic development. This model is called the “Two-gap model” and has been discussed extensively in the context of foreign aid effectiveness. Therefore, the discussion in this section could reflect this point.

Thank you. This was incorporated in the literature review section.

  1. L619- Table 3. Summary of Variable Description
    Specify the unit of RR, L and IMP (in USD?). Check whether the variables used in the model are the constant price term
    .

Yes they are in constant price terms.

Round 2

Reviewer 2 Report

Thank you for your new version of the maniscprit. Now I have no futher suggestions about revision. 

Reviewer 3 Report

I reccomend the paper to accept in present form.

This manuscript is a resubmission of an earlier submission. The following is a list of the peer review reports and author responses from that submission.

Round 1

Reviewer 1 Report

1.The author chosen chin's model try to explain how  chinese loan   to impact africa contries. in this study, it is a weak negative relationship was discussed. Please strengthen the explanation of this part and echo the research results with the past literature.

2.The hypothesis of the research is supported by the literature. The author does not infer the formation of various hypotheses, but infers the results between the two from the official data, which is very weak. It is suggested that the author infer the formation of hypotheses in each paragraph of section 2.

3.It is suggested that the author should ask the MDPI editing center to revise the paper and submit it again.

Reviewer 2 Report

Summary: The impact of China's lending to African countries has been a controversial topic. On the one hand, supporters believe that these loans help Africa build infrastructures that promote local economic growth. On the other hand, more critics claim that China loans to projects with a weak linkage to development, and these loans could drag African countries into debt traps. In this paper, the author provides new evidence to this debate by examining the impact of China's loans on economic growth in African countries using annual panel data from 15 African countries. By estimating a Mean Group and a Pooled Mean Group model, the author shows that China's infrastructure loans are beneficial to Africa since they lead to economic growth in African countries.

Overall, I think the development impact of infrastructure is a very important and interesting topic. There is rich literature showing that large infrastructure projects such as electricity and roads substantially affect structural transformation or economic growth in developing countries. But their economic impacts in African countries are still under debate. This paper is new in that it provides evidence to the effect of infrastructure loans on GDP growth in African countries. However, I think this empirical evidence is still preliminary.  My comments are as follow:

  1. In Section 1, the background information takes too many spaces while the introduction of the paper is somewhat neglected. As a reader, I would expect to see a brief introduction of the question this paper hopes to answer, how the authors will answer it, and what they find. As to the background information, it would be better to discuss China's development finance to Africa briefly and its critics, then move detailed information to Section 3.
  2. Although I like the detailed information displayed in Section 3, most of it is a narrative description, making it less like an economic paper. For example, I believe those views listed in Table 1 can show the benefits of China's activities in Africa, but these are simply views. I doubt the necessity of highlighting them in an economic paper. In addition, the comparison in Table 2 is a bit abstract. It would be better to give some specific numbers. For example, how low are China's interest rates compared with the other two sources? In addition, since we're comparing three financing sources, can you provide information about how the share of three sources varies over time? This will give a better sense of the importance of china's loans. Similarly, there is no need to discuss details of critics about China's loans, such as vanity projects, predatory loans, and control and dominance. A few sentences would be adequate.
  3. The data description is a bit short. A summary statistic of the main variables in the main equation would be good for readers. Personally, I would also like to see a summary of China's loans to the 15 countries included in the paper.
  4. The independent variables in Eq.(1) include resources rents (RR), China's loans (L), total imports (IMP), FDI and value-added of agricultural sector (AGR). It's easy to understand the term L, but the inclusion of other terms requires some justification. In particular, why imports not exports? How does FDI differ from loans? Another concern is that GDP may not be a good indicator of economic activity and growth effects in the presence of foreign aid. I would recommend considering more dependent variables such as government revenues.
  5. In presenting the results, I like the comparison of MG and PMG estimators, and the short-run and the long-run results. But it's not clear how the short and long-run effects are measured.
  6. In Eq.(1) and presenting the results, the importance of L is not highlighted among other terms. Without knowing the topic, it's easy to identify this paper is about the impact of resources rents, imports,...
  7. More importantly, this paper implies that China's loans promote GDP growth because they provide Africa with needed infrastructures. But the variable L is just a rough indicator for the development of local infrastructure. Moreover, given how GDP is calculated, a concern is that GDP grows simply because more loans bring more projects under construction for that year. Thus, the GDP growth could be fueled by increasing loans instead of better infrastructure. If China's loans cannot effectively improve infrastructure in African countries, the loans could eventually drag these countries into debt traps. This might not be the case. But the paper fails to eliminate this possibility. Hence, I recommend the author provide evidence that China's loans go to investments in local infrastructure. For example, the author could report project-level data about each loan.

Reviewer 3 Report

The paper attempts to analyze a widely discussed topic concerning China’s contributions to African countries, but it fails to present literature review, research methodology and policy implications properly.

Suggestions:
1. Literature review:  this part needs a significant improvement on the logic linkage and relevance among the cited literature. The current literature review does not demonstrate an adequate understanding of the relevant literature in the field, especially the theories of international loan. Also, the novelty of this paper needs to be justified in this part.

2. Hypothesis and Methodology: I don’t see the hypotheses in the paper, even those in an implicit way. I suggest the addition of that. The methods actually used are not consistent with that claimed (GMM) in the abstract. The use of the methodology should be on the grounds of data and variables, instead of the previous studies, as argued by the paper “The study chose Chin et al. (2021) model because their study at- 411 tempted to answer the same question that this study seeks to answer.” Why is the model by Chin et al. (2021) certainly appropriate for this topic? It is not a convincing argument.

3.Conclusions:  the paper concludes with some points which are not supported by the analysis, for example, the summary in the second paragraph of the conclusion part is not consistent with the findings. Moreover, the implications for research and practice are not well identified.

4.Quality of Communication:  the paper has not been edited adequately and needs improvement. Some syntax, phrase and sentence errors are present.

Reviewer 4 Report

REVIEW COMMENTS: CHINA IN AFRICA: AN EXAMINATION OF THE IMPACT OF CHINA’S LOANS ON GROWTH IN SELECTED AFRICAN STATES

Comment

This is potentially an interesting subject to explore. This paper if well improved, has merit and would contribute to both scholarship and practice. The areas that need massive improvement are the introduction, literature, methodology, and argumentation. Kindly address the questions raised in the attached annotated manuscript and in my comments below.

TITLE

A juxtaposition of the manuscript’s title and aims raises ambiguity: I suggest the title be changed to “China in Africa: An Examination of the Impact of China’s Infrastructural Loans on Growth in Selected African States.”

ABSTRACT

The study claims to use GMM panel techniques, which is not the case.

INTRODUCTION

Rather than wait to the end to state the aim of the paper, it is imperative to avoid testing the patience of the reader. Authors are advised to state the aim of the paper beforehand. Authors can state the focus and empirical results of extant literature on the subject matter and how innovative the findings of the present study are. Also, authors fail to discuss the stylized facts of China’s infrastructural loans to Africa.

LITERATURE

Literature needs complete overhaul and revamping. Instead of focusing on literature on the effect of infrastructural loans on economic growth, authors to a large extent, take a swipe to focus their attention on literature on effect of infrastructure on economic growth. Additionally, authors can discuss the literature on the effect of infrastructural loans on economic growth (which is very paramount because of the aim of the study before discussing the effect of infrastructure on economic growth.

METHODOLOGY

  1. Authors should highlight the reasons why the scope of the study was limited to debt-stressed African countries.
  2. Authors must justify the selection of the variables used in the model. What is the underlying reason for including imports (IMP) and omitting exports?
  3. Also, economic growth models usually include variables such as capital formation, government expenditure, inflation or exchange rates and institution variables, however this present study omits all of these variables in its model and empirical analysis.
  4. I suggest authors refer to the following articles and related others:
  • Cudjoe, D.A., Yumei, H. and Hu, H. (2021), "The impact of China's trade, aid and FDI on African economies", International Journal of Emerging Markets, Vol. ahead-of-print No. ahead-of-print. https://doi.org/10.1108/IJOEM-10-2020-1180
  • Iyke, B.N. (2017), “Does trade openness matter for economic growth in the CEE countries?”, Review of Economic Perspectives, Vol. 17 No. 1, pp. 3-24, doi: 10.1515/revecp-2017-0001
  • Busse, M., Erdogan, C. and Muhlen, H. (2016), € “China’s impact on Africa – the role of trade and FDI”, Kyklos, Vol. 69 No. 2, pp. 228-262, doi: 10.1111/kykl.12110
  • Agbloyor, E.K., Gyeke-Dako, A., Kuipo, R. and Abor, Y.J. (2016), “Foreign direct investment and economic growth in SSA: the role of institutions”, Thunderbird International Business Review, Vol. 58 No. 5, pp. 479-497, doi: 10.1002/tie.21791
  • Fayissa, B. and Nsiah, C. (2013), “The impact of governance on economic growth in Africa”, The Journal of Developing Areas, Vol. 47 No. 1, pp. 91-108, doi: 10.1353/jda.2013.0009.
  1. In the variable denotation, authors fail to specify the dependent variable, independent variables and control variables and leaves it for readers to make implication.
  2. In particular, without scaling some of the key dependent variables--e.g., Chinese Loans to African Governments, Total imports of goods and services, Total natural resources rents FDI, the results are totally meaningless, regardless of the econometric method used. This is for the obvious reason that a given amount of Chinese Loans, or FDI will have an impact on a country's GDP growth if it is a significant share of the GDP to begin with. The scaling will also help reduce the overall variability within the data and to avoid any potential bias.

ANALYSIS

  1. The results are presented clearly. However, I believe the discussion can be significantly improved.
  2. The study fails to explain the mechanism under which imports lead to economic growth in Africa.
  3. The study fails to analyze the results of some variables.
  4. The manuscript misses the implications for research, practice and/or society. It does not identify clearly any implications for research, practice and/or society. Also, it fails to bridge the gap between theory and practice.
  5. The quality of communication is not high enough. There are occasional awkward turns of phrases and grammatical errors. I have highlighted some of them in the attached peer review manuscript. Overall, the exposition needs improvement.
  6. Authors fail to present the summary statistics for the variables.
